# Holistic Evaluation of Digital Applications in the Energy Sector—Evaluation Framework Development and Application to the Use Case Smart Meter Roll-Out

**Paul Weigel [1,2,\*], Manfred Fischedick [1] and Peter Viebahn [1]**

[1] Wuppertal Institute for Climate, Environment and Energy, Döppersberg 19, 42103 Wuppertal, Germany; manfred.fischedick@wupperinst.org (M.F.); peter.viebahn@wupperinst.org (P.V.)
[2] Institute for Environmental Process Engineering and Plant Design, University Duisburg-Essen, Leimkugelstr. 10, 45141 Essen, Germany
[\*] Correspondence: paul.weigel@uni-due.de

**Abstract:** The development of digital technologies is accelerating, enabling increasingly profound changes in increasingly short time periods. The changes affect almost all areas of the economy as well as society. The energy sector has already seen some effects of digitalization, but more drastic changes are expected in the next decades. Besides the very positive impacts on costs, system stability, and environmental effects, potential obstacles and risks need to be addressed to ensure that advantages can be exploited while adverse effects are avoided. A good understanding of available and future digital applications from different stakeholders' perspectives is necessary. This study proposes a framework for the holistic evaluation of digital applications in the energy sector. The framework consists of a combination of well-established methods, namely the multi-criteria analysis (MCA), the life cycle assessment (LCA), and expert interviews. The objective is to create transparency on benefits, obstacles, and risks as a basis for societal and political discussions and to supply the necessary information for the sustainable development and implementation of digital applications. The novelty of the proposed framework is the specific combination of the three methods and its setup to enable sound applicability to the wide variety of digital applications in the energy sector. The framework is tested subsequently on the example of the German smart meter roll-out. The results reveal that, on the one hand, the smart meter roll-out clearly offers the potential to increase the system stability and decrease the carbon emission intensity of the energy system. Therefore, the overall evaluation from an environmental perspective is positive. However, on the other hand, close attention needs to be paid to the required implementation and operational effort, the IT (information technology) and data security, the added value for the user, the social acceptance, and the realization of energy savings. Therefore, the energy utility perspective in particular results in an overall negative evaluation. Several areas with a need for action are identified. Overall, the proposed framework proves to be suitable for the holistic evaluation of this digital application.

**Keywords:** digitalization; digital applications; cyber–physical systems; energy sector; sustainability; holistic evaluation; framework; multi-criteria analysis; life cycle assessment; expert interviews; smart meter

## 1. Introduction

Digitalization is often seen as a megatrend of our time. However, this trend began several decades ago with the use of the first industrial computers. Nevertheless, due to the exponentially accelerating developments of digital technology and its transformative speed and impact, digitalization is now one of the main drivers for change in our industry and society. The technological developments are often mutually reinforcing; thus, the trend is expected to continue to accelerate. This fundamental socio-economic transformation is envisioned in [1] and described from an economic perspective in [2]. Some areas of economy

and society, such as commerce, banking, and communication, have already undergone significant changes and are today the digital leaders, while others remain mostly unchanged as identified in a survey among German companies [3]. In fact, the energy sector was a pioneer of digitalization in using early digital monitoring and control systems in power plants and the transmission network, as explained in the educational publication on power supply systems [4], decades before the term digitalization became widely used. Today, many digital solutions have already been implemented on all energy value stream steps, yet the value stream itself remains mostly unchanged. Looking forward, a wide variety of future potential digital applications can already be identified. An overview of applications is presented in [5]. Numerous publications describe specific applications, e.g., a concept of universal smart machines for the energy sector consisting of electric protection equipment, and electricity quality measuring and monitoring devices is recently presented in [6]. These applications offer enormous potential to positively impact all three aspects of the energy sector "target triangle" (cost, system stability, environmental impact) and might even affect the value stream itself. Whether or not the energy value stream fundamentally changes, digitalization will likely play an essential role in the changes of the energy sector within the next decades [7].

For companies, regulators, and non-governmental stakeholders, a good understanding of the developments, the digital applications available now and in the future, and their associated impacts, benefits, and risks is essential. On the one hand, this transparency is needed for broad societal and political discussions about the targets and pathways of the digitalization. On the other hand, it is an essential piece of information for companies to develop and sustainably implement digital applications.

Therefore, a framework suitable for the evaluation of digital applications in the energy system is required. Digital applications in the energy sector can be extremely diverse (see definition in Appendix A) and a large number of stakeholders are potentially involved. A broad range of potential technical, economic, ecological, and social impacts can be caused [5]. In particular, ecological impacts can be quite substantial and often require a lifecycle perspective for full coverage. Furthermore, the development of new applications can be rapid. Due to the described nature of digital applications, an evaluation framework needs to be highly flexible, on the one hand, yet provide sufficient guidance to make alternatives comparable on the other. It needs to cover the wide range of potential impacts and be able to cope with qualitative as well as quantitative evaluation criteria. The integration of a life cycle perspective must be possible, at least for parts of the evaluation. Furthermore, the perspectives of different stakeholders need to be integrated. Besides that, the level of detail of the assessment needs to be flexible to cope with the varying availability of data and data of varying quality. The flexibility on the level of detail is also important to allow the user to decide if an initial "Quick Check" is desired or rather a "Deep Analysis".

A variety of methods are applied in the assessment and evaluation of technologies, for example, the technology assessment (TA), the cost benefit analysis (CBA), the multi criteria analysis (MCA), and the life cycle assessment (LCA). In the following paragraphs, we analyze whether and how these methods could be used for the intended purpose.

The technological assessment (TA) is a tool to systematically study unintended, indirect, or delayed impacts of technological developments on society [8]. In general, the TA is open to different ways of assessing the content matter; however, it provides guidelines regarding the overall process. Several types of TA are discussed in the edited volume [9]. The advantage of the TA is its very flexible approach, which can be applied to any technology and can cover all impacts. The disadvantage is that no specific methods for assessments are given such that the result heavily depends on the skill and experience of the person/organization conducting the TA, e.g., ref. [10] states that one success factor is to ensure dedicated academic capacity. Although the method gained high popularity already in the 1970s when the U.S. Office of Technology Assessment was established, it is still frequently applied today in the context of supporting public and parliament discourses, and continues to be refined for this purpose as, for example, in [11]. It is also applied in the field of

digital technologies and in the energy sector. An application of the method for theich isture of t of the digitalizationsocialheate than the expert interviews.ange the tendency of the rsult.e aspects of grid and assessment of "clean energy" is, for example, presented by the International Energy Agency (IEA) in [12]. In [13], the method was recently used to assess the future of industry 4.0, which is a key part of the digitalization, and in [14], TA is used to identify and evaluate digitalization measures for battery cell production. Due to the open nature of the approach, however, TA does not give sufficient guidance on which assessment tools to use and which criteria to consider for the intended purpose of a digital application evaluation framework.

The cost benefit analysis (CBA) is based on the monetary value of efforts and impacts and makes the monetary implications of different alternatives comparable. Non-quantifiable effects can only be included informatively. Although, in theory, a holistic approach is possible, in some assessment areas, such as social acceptance or human safety, it is more difficult to calculate the monetary value as the authors explain in [15]. CBA guidelines for specific applications, including digitalization topics, are provided by the European Commission and are publicly accessible. For example, ref. [16] gives guidance on performing a CBA for smart meters, ref. [17] for smart grids, and [18] for investment projects. Examples of the use of CBAs in the energy industry are numerous. In the context of this study especially, the CBAs focusing on smart meters are relevant. Reference [19] presents a CBA for the German smart meter roll-out, ref. [20,21] provide CBAs for the roll-out in Great Britain, and [22] is a review study of the different CBAs performed of smart meter roll-outs in Europe. Of course, the CBA method can also be applied to other topics in the energy sector, such as energy planning [23] and, more recently, the evaluation of digital condition monitoring for remote maintenance systems in power plants [24]. Due to the limitation to impacts, which can be quantified as monetary values, however, the CBA is not suitable for holistically evaluating digital applications.

The multi criteria analysis (MCA) method consists of a range of different impact methods, which all use multiple criteria to evaluate alternatives and support the decision-making process, among them being multi attribute utility and value theories (MAUT and MAVT), simple additive weighting (SAW), and the analytical hierarchy process (AHP), as well as outranking methods. A short and concise overview including the advantages and disadvantages per method is presented in [25]. MCA methods are especially suitable for complex subject matters with multiple objectives and multiple stakeholders with different perspectives. It is possible to integrate qualitative as well as quantitative criteria and take the entire life cycle into account [26]. The MCA method has been used in the energy sector for several decades and has become even more popular in recent years [27]. An extensive literature review regarding MCA methods, the area of application, the stepwise approach, strengths and weaknesses as well as used criteria is performed in [28]. The context of the presented literature review is the sustainability assessment of renewable energy development. The authors conclude that no single impact method can be identified as the best or worst and that hybrid methods, therefore, are becoming increasingly applied. A recent practical example of the use of MCA methods in the energy sector is [29]. Here, multi-actor adoption of the MCA is used to evaluate different energy options for a German village. Multi-actor adoption allows for the determination of stakeholder-specific criteria, besides stakeholder-specific weights. A recent non-energy-related example of an MCA application is performed in [30]. Here, a multiple criteria approach is used to assess investment projects, including risk factors where different risk components are included as criteria. Furthermore, the MCA is also already applied to assess specific topics of the digitalization in the energy sector. In [31], the AHP is used to assess digital energy services regarding energy efficiency. In [32], an MCA approach is used to identify the most beneficial robotics technology for the power industry. Due to its well-proven use for energy and digital topics and the possibility to include perspectives of various stakeholders and qualitative as well as quantitative data, the MCA is found to provide a suitable basis for the intended digital application evaluation framework.

The life cycle assessment (LCA) follows a cradle-to-grave approach with a focus on identifying and quantifying environmental impacts [33]. Depending on the quality of the input information, it can provide precise information as to where along the life cycle environmental impacts occur. The LCA is defined by the standards in [34,35]. Guidelines for the application of the method are given in [36,37]. In a broader sense, the life cycle approach can be used for a life cycle sustainability assessment (LCSA) as a combination of LCA, life cycle costing (LCC), and social life cycle assessment (S-LCA) [38]. Two LCA approaches can be distinguished: attributional and consequential LCA. While the attributional LCA looks at impacts directly attributed to a product/service, the consequential LCA also takes into account the consequence (marginal effects) in the wider system caused by the product/service (e.g., change of use patterns) as stated in [33]. Tools and databases are publicly and commercially available and make the LCA easily applicable. However, the skill and experience of the practitioner has a direct effect on the modeling as stated by [39]; therefore, in-depth knowledge about the assessed life cycle as well as the analysis methods are required to yield meaningful results. Applications of LCA methods in the energy sector are very common. The energy sector is the main contributor to environmental effects, such as the emission of GHG [40]; therefore, a detailed understanding of the impacts along the life cycle in the energy sector is of outstanding importance. A literature review of LCA studies regarding electricity generation technologies is published in [41]. An assessment of LCA applications to assess GHG removal technologies is recently discussed in [42]. The LCA of the deployment of smart meters in the Californian energy system, presented in [43], is frequently referenced in this paper. LCA methods have already been applied to a variety of digital topics. An assessment of different home energy management applications is presented in [44]. In [45], a LCA concept for assessing new industry 4.0 applications is proposed, which uses ICT to automatically gather and monitor relevant mass and energy flows. Overall, the LCA is a well-established method for assessments with a life-cycle perspective of energy and digital topics and enables the inclusion of effects along the entire life cycle. Therefore, it is found to be a suitable assessment tool to be used in the context of the proposed framework.

Some of the methods described above can be combined. A recent example of the combination of MCA and LCA is [46], wherein a surface treatment process is evaluated. Eight criteria in the categories of economy, ecology, and technology are assessed in an LCA. The results are aggregated, using three different MCA methods. All criteria are quantitative. The authors conclude that the combination of MCA and LCA has a high potential while not finding any significant differences in the outcome between the three MCA methods. The prosuite project, documented in [47,48], aimed at developing a new methodology for the sustainability impact assessment of new technologies. A combination of LCA and MCA is proposed. The authors suggest a new hierarchical structure of impact categories, criteria, and so-called contributors. A LCA is used to assess the criteria. All criteria are quantitative. For the aggregation, the MCA methods of weighted sum and outranking are used. The project shows how LCA and MCA can be combined as a powerful assessment and evaluation tool. In the energy sector, a combination of the LCA and MCA methods is used by [49] to assess and evaluate 11 energy mix scenarios for Mexico until 2050, along 17 criteria in the categories of environment, economy, and society. The combination of MCA and LCA is well established. However, without further assessment methods, the combination lacks the possibility to cope with non-quantifiable data and is, therefore, not suitable for the intended purpose.

Furthermore, the MCA is also commonly combined with stakeholder/expert interactions. A recent example in the energy sector is [50], where a participatory MCA is conducted (and combined with system modeling) to assess linkages between water resources and energy technologies in Morocco and evaluate water-saving measures. Digitalization topics have also already been evaluated based on a combination of MCA and stakeholder/expert interactions. In [51], most recently, the authors use an MCA method as well as expert interactions to identify the relevance for the sustainability of different enablers of the

industry 4.0 development. An approach to assess countries' industry 4.0 readiness level based on MCA methods and expert interactions is recently presented in [52]. The assessment via expert interactions is suitable for qualitative information; however, results may have higher uncertainty. In the context of the evaluation of digital applications in the energy sector, the long-term environmental impacts, in particular, require a more thorough life-cycle assessment.

In [53], the authors conduct an MCA to compare electricity generation technologies combining the MCA with both LCAs (for qualitative data) and expert judgment (for qualitative data). For each criterion, a specific assessment method, i.e., LCA or expert judgment, is determined. This approach offers many of the characteristics required for the intended purpose of the study presented here. However, the clear allocation of assessment methods to criteria does not provide sufficient flexibility for the purpose of evaluating the wide variety of digital applications with different availabilities of data.

As revealed by the discussed literature, the application of evaluation methods in the energy sector is frequently used and well-proven, especially the combination of MCA and LCA. Besides that, first applications of the discussed evaluation methods for specific digital topics are identified. None of the discussed methods or combinations of methods, however, are directly suitable as an evaluation framework for the variety of digital applications in the energy sector. Although the approach presented in [53] already shows several important characteristics, it does not meet the required specifications described above. Furthermore, the promising combination of MCA, LCA, and expert interactions has not yet been applied to digital applications. Therefore, the aim of this paper is to provide an evaluation framework for digital applications with the required properties and to demonstrate it on a selected example.

The remainder of the article is organized as follows: the proposed evaluation framework and its adaption are described in Section 2. In Section 3, the application to a German smart meter roll-out is evaluated as a first test of the framework's suitability. Finally, the results of the evaluation are discussed in Section 4, whilst a critical reflection of the suitability of the proposed framework is performed and potential adaptions are suggested in Section 5.

## 2. Proposed Evaluation Framework

Overall, the discussed literature offers a sound basis for the development of the proposed evaluation framework to close the identified gap. Going beyond the existing approaches discussed above, this paper develops an approach providing novelties on three levels:

(1) A framework is proposed based on three well-established methods, namely the multi-criteria analysis, the life-cycle assessment (consequential LCA in this case) and expert interviews. The methods can be combined in a flexible way depending on the desired level of detail and the availability of data.

(2) Since the evaluation of digital applications in the energy sector is the focus of the framework, it is adapted to the special characteristics of these applications. This is mainly done by providing a set of criteria and weighting profiles.

(3) In order to validate the theoretical concept, the framework is applied to the use case of the German smart meter roll-out.

The objective of the proposed framework is to provide a structured and transparent basis for the holistic evaluation of digital applications in the energy sector. The result of the holistic evaluation can facilitate broader societal and political discussion and support companies and other stakeholders to identify risks and critical aspects as well as potential solutions. The intended user of this evaluation framework are fellow researchers, companies of the energy sector, and governmental and non-governmental organizations.

*2.1. Framework Development by Combination of Three Methods*

As depicted in Figure 1, the main evaluation methodology of the framework is a MCA, providing the overall evaluation structure. The assessment of the different criteria defined in the MCA, i.e., gathering and assessing the information, is conducted either within the LCA for quantitative aspects or via the expert interviews (EI) for (mostly) qualitative aspects. This combination of the LCA and EI offers flexibility to the user to choose the desired level of detail. Either a qualitative "Quick Check", encompassing the four assessment dimensions but based only on expert interviews, or a "Deep Analysis" based on detailed LCA results besides the expert interviews can be performed. Due to the high impact of the energy sector on the environment, it is of outstanding importance to understand the full environmental impacts of changes caused by digital applications. To cover all environmental impacts, a life-cycle perspective is necessary. Therefore, the deep analysis, based on an LCA, is applied for the assessment of the ecological criteria.

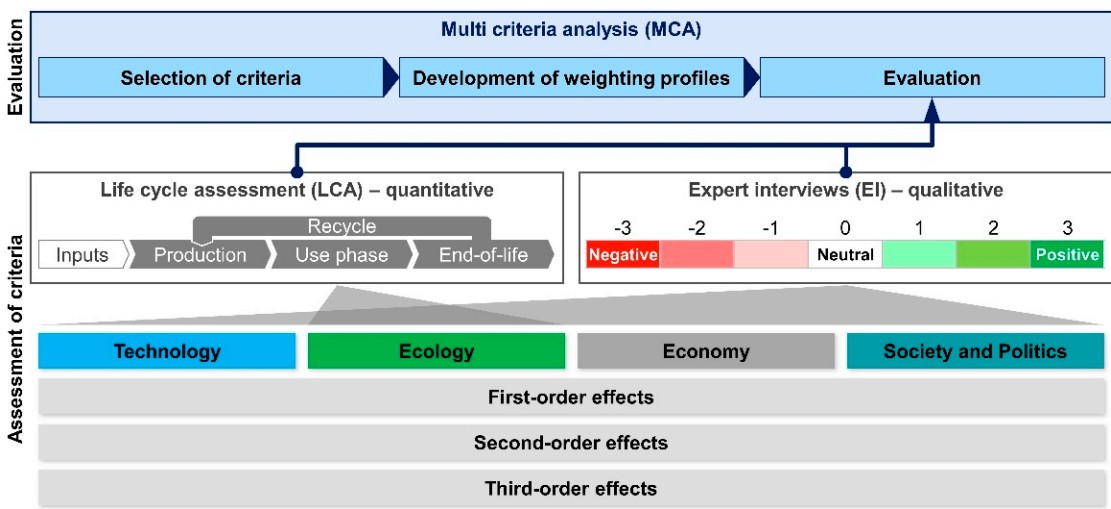

**Figure 1.** Framework for holistic assessment and evaluation of digital applications based on [5].

The MCA offers the possibility to aggregate the results on different levels, enabling a strong reduction of complexity on the one hand and providing relevant detailed information for in-depth discussions on the other. Based on the intended receiver of the result (e.g., scientific researcher, employee, or politician), an adequate level of detail can be chosen.

2.1.1. Multi-Criteria Analysis (MCA)

The general MCA approach [54] is slightly modified. The following steps are performed:
1. Definition of the evaluation subject(s).
2. Identification of the relevant evaluation criteria.
3. Determination of the weighting of the criteria.
4. Assessment and evaluation of the criteria.
5. Aggregation of results and comparison of the alternatives.

As the proposed framework is intended to be suitable for all digital applications in the energy sector, step 1, the definition of the evaluation subject(s) is performed once to collectively define digital applications and derive necessary evaluation characteristics, and subsequently for each individual digital application under evaluation.

In order to achieve a holistic perspective, the evaluation criteria identified in step 2 belong to four categories—technology, ecology, economy, and society/politics (Section 2.2). These criteria cover most possible impact areas. Therefore, the framework is suitable for diverse kinds of existing digital applications as well as potential new applications. In the case that an impact is not reflected, the approach can be adapted by adding the respective criteria.

Step 3 includes the definition of weighting factors per criteria based on expert interviews. The perspectives of multiple stakeholders can be integrated via different weighting profiles (Section 2.3). The proposed weighting profiles are seen as a proposal and can be adapted as needed by the user of the framework.

The assessment of the criteria is conducted via the LCA and/or EI. The evaluation range is defined to be −3 to +3, with −3 being a very negative impact and +3 being a very positive impact. The evaluation range is depicted in Figure 2.

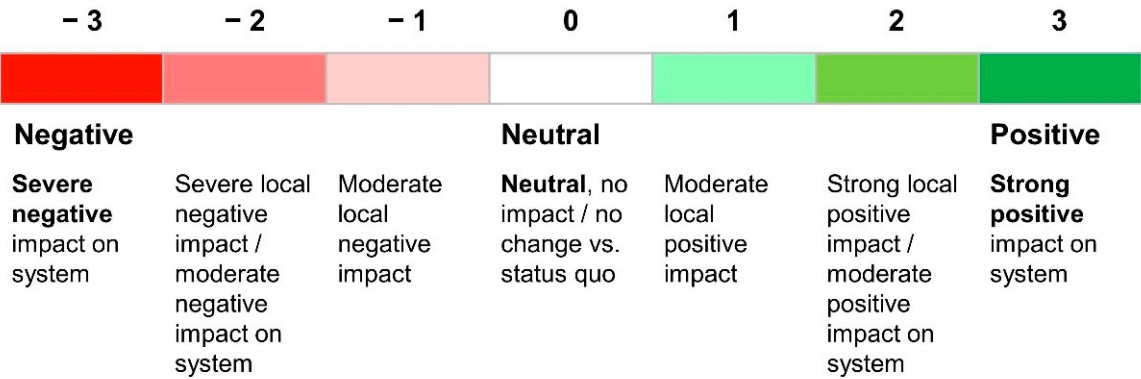

**Figure 2.** Evaluation range for MCA criteria in the case of expert interviews.

After the assessment of the criteria, the results are aggregated (Section 2.4). The compensatory system allows for the aggregation of the results of individual criteria into overall scores. Full aggregation (i.e., compensation) is possible, yet the user may decide to display the results on different aggregation levels, thus providing more detail.

2.1.2. Life Cycle Assessment (LCA)

According to [34], the LCA generally consists of the following four steps, which can be performed iteratively:

1. Goal and scope definition.
2. Inventory analysis.
3. Impact assessment.
4. Interpretation of the result.

The definition of the goal, scope, and functional unit in step 1 needs to be in line with the overall definition of the evaluation subject of the MCA, as the methods are combined in the framework. In particular, it needs to be considered whether an isolated application (attributional LCA) or the application, including system-wide effects (consequential LCA), is assessed. In the context of the proposed framework, in most cases, the consequential LCA is more appropriate since not only the application of one single product itself, but the impacts caused by a nationwide application are of interest. Steps 2 and 3 follow the general LCA approach. Step 4, the interpretation of results, is subsequently performed in the MCA as part of the overall framework.

2.1.3. Expert Interviews (EI)

The EI is a method used to quickly gather high-density information if a mostly qualitative and subjective result is acceptable.

For this study, semi-standardized systemizing interviews, as described in the educational edited volumes ([55], p. 33) and ([56], p. 465), are conducted, using the list of criteria as a structure. This offers a good compromise between comparability between the information gathered in different interviews and ensuring that the relevant aspects are identified and covered. To minimize the interviewer's influence on the outcome, an approach of minimal interventions is used, meaning that after an initial introduction and explanation, no input from the interviewer is given throughout the central part of the interview as long

as the interview partner does not raise any questions or the interviewer identifies any misunderstandings. The interviews can include quantifiable and non-quantifiable first-, second-, and third-order effects, thus all criteria can be assessed. However, it needs to be noted that EIs, unless conducted in a fully standardized way with a high number of experts, cannot be representative. However, as the expert is seen as a representative for a larger group, reasonable results can be expected.

*2.2. Criteria Selection*

The selection of criteria is of particular importance, as it defines the range of effects considered in the analysis. The criteria need to be MECE (Mutually Exclusive and Collectively Exhaustive: No Overlap/Duplication, Fully Comprehensive) as well as relevant for the evaluated alternatives. In particular, it must be ensured that all stakeholders' perspectives are reflected.

The suggested list of evaluation criteria is based on an analysis of the existing literature, the corresponding author's own professional experience in the energy sector and expert interviews. In the first step, a long list of criteria is gathered based on the existing literature. MCA review papers, as well as articles on specific MCA applications, are analyzed. The primary focus is on MCAs in the energy sector. In particular, the publications [28,57–60] are considered. The broadest overview of used criteria, based itself on a very comprehensive literature review, is presented in [59]. The publications reviewed in [59] include a variety of different technologies as well as different analysis objectives. Therefore, the identified "typical" evaluation criteria are relatively high level and holistically cover the areas of technology, economy, ecology, and society. A similar structure of criteria categories is also found in several publications reviewed in [28]. Two specific MCA applications used to develop the criteria list are [58,60]. In [58], the case of regional use of bioenergy in Germany is used to demonstrate the decision-making process based on the MCA. Although the criteria are not structured into the four previously mentioned categories, collectively, they do cover these categories. The publication [60] adds the category legislation to the four categories for the MCA of bioenergy options in Tanzania.

The resulting long list of criteria is subsequently extended and structured based on the corresponding author's own professional experience of several years in the energy industry and with digitalization endeavors. Lastly, the criteria are refined and validated in expert interviews. Several interviews are conducted with a variety of experts, including one expert for IT security, two university professors in the field of power engineering, one representative of an energy distribution network company, one management consultancy executive with a specialization in the energy sector, one member of the energy department of the German consumer protection organization and one researcher in the field of energy technologies and life-cycle assessment. Overall, the technical, ecological, economic as well as societal perspectives are covered via the selection of experts. The interviews are conducted sequentially and on the basis of the long list of the previously identified criteria. In some cases, changes made in a later interview have to be reiterated with experts of previous interviews. Although a correlation or an overlap between criteria should be avoided, this is not possible in all cases. In particular, some technical criteria also have an impact on ecologic, economic, and societal aspects. However, these overlaps are minimized.

Overall, this approach resulted in the list of criteria displayed in Table 1. The list is structured into 4 + 1 categories, including the technology, ecology, economy and society/politics categories as well as a cross-functional risk category. The cross-functional risk of failure category consists of one criterion regarding the probability of failure and four criteria covering the impacts of failure per each of the four other categories. For each application, a failure scenario needs to be defined.

**Table 1.** List of MCA criteria.

| Technology (hard and software) | | Economy | |
|---|---|---|---|
| Generation controllability | resulting effect | Profitability for supplier | resulting effect |
| Operating reserve | resulting effect | CAPEX for supplier | effort |
| Ramp-up/down speed | resulting effect | OPEX for supplier | effort |
| Plannability of generation/consumption | resulting effect | Revenue for supplier | resulting effect |
| Black start ability | resulting effect | Amortization period | effort |
| Network controllability | resulting effect | Strategic advantages/disadvantages for supplier | resulting effect |
| Power flow optimization (high voltage) | resulting effect | Accessing new customer segments | resulting effect |
| Regional/local balancing (medium/low voltage) | resulting effect | Global growth potential | resulting effect |
| Energy savings/demand | resulting effect | First mover advantage/ disadvantage | resulting effect |
| Power demand | resulting effect | Further economic effects | resulting effect |
| Gas / heat demand (only if relevant) | resulting effect | Competitive situation | effort |
| Vulnerability of critical infrastructure | resulting effect | Market entry barriers | effort |
| Dangers of cyber attacks | resulting effect | Profitability for users | resulting effect |
| Dependence on IT systems | resulting effect | CAPEX for users (only if provider $\neq$ user) | effort |
| Security of private and company data | resulting effect | OPEX for users (only if provider $\neq$ user) | effort |
| Confidentiality of private and company data | resulting effect | Revenue for users (only if provider $\neq$ user) | resulting effect |
| Integrity of private and company data | resulting effect | Amortization period | effort |
| Availability of private and company data | resulting effect | Non-monetary benefits for users | resulting effect |
| Technological marketability (market readiness) | effort | Comfort (Convenience) | resulting effect |
| Technical implementation effort | effort | Usability | resulting effect |
| Hardware and constructions | effort | Transparency and controllability | resulting effect |
| Software | effort | Economic security and independence | resulting effect |
| Technical operating effort | effort | **Society and Politics** | |
| Technical disposal / recycling effort | effort | Social acceptance / rejection | effort |
| Availabilities of materials, know-how and capacities | effort | National added value | resulting effect |

**Table 1.** *Cont.*

| | | | |
|---|---|---|---|
| Availability of raw materials | effort | National value creation steps | resulting effect |
| Availability of know-how and capacity for production | effort | Jobs | resulting effect |
| Availability of know-how and capacity for inst. / O&M | effort | Working conditions | resulting effect |
| Availability of know-how and capacity for recycling/disposal | effort | Privacy | resulting effect |
| Innovation potential with/without retrofit possibility | resulting effect | Participation in the energy sector | resulting effect |
| Interdependencies (synergies/competition) | resulting effect | Generational justice | resulting effect |
| Interdependencies with applications in the energy sector | resulting effect | Dependence on other nations | resulting effect |
| Interdependencies with applications in other industries | resulting effect | Regulatory implementation effort | effort |
| **Ecology** | | Governmental support | effort |
| Environmental effects | resulting effect | **Probability and impact of failure** | |
| Energy balance | resulting effect | Probability of failure | risk |
| Greenhouse gas emissions | resulting effect | Technical impact | risk |
| Resource depletion | resulting effect | Ecological impact | risk |
| Human toxicity | resulting effect | Economic impact | risk |
| Enabling integration of renewable energies | resulting effect | Societal impact | risk |

*2.3. Criteria Weighting*

The weighting of the criteria is necessary to reflect differences in the importance of different criteria. Weighting becomes especially important when the multi criteria analysis includes the perspectives of different stakeholders. Either one weighting profile, which covers all stakeholders' perspectives, can be defined, or the stakeholders' different views can be represented in different weighting profiles.

An extensive range of methods is available to determine weighting factors, each with inherent advantages and disadvantages. The method of criteria weighing might have a stronger effect on the overall result than the method of the MCA as suggested by [61]. A comprehensive overview of weighting methods is given in ([62], pp. 46–58), as the authors extend the previously prepared review by [63]. The authors conclude that there is not one method that proves to be superior but the method rather needs to be chosen individually based on the MCA method used, the type of criteria, the available information, and the knowledge and skill of the person determining the weighs. Overall methods can be classified into direct vs. indirect, objective vs. subjective [62,64], and compensatory vs. non-compensatory [65]. Well-known and broadly used weighting methods include ranking, rating (including point allocation), and pairwise comparison (including the analytical hierarchy process).

In the context of this study, point allocation, a direct, subjective, and non-compensatory rating method, is used. This method bears the inherent advantage of its simplicity and is found to be well working in the experimental comparison of weighting approaches in [66]. It is chosen to use 100 allocation points. The 100 points represent 100% importance, a concept that is understood by experts and decision makers without further explanation. Once all points are allocated, in order to increase the weight of one criterion, the point allocator needs to reduce the weight of another. This forces the point allocator to thoroughly question previously determined weights. In this case, after the allocation of all points, a sense check is encouraged by comparing selected pairs of criteria in an iterative process to ensure that both the weighting order and the weighting distances between them reflect the allocator's preference. This sense check is necessary, due to the large number of criteria. As the approach starts with weighting the high-level criteria categories, the number of criteria within each category does not influence the overall weight of the category. The risk of non-matching criteria units is encountered by choosing matching units and communicating those to the point allocators.

The weighting itself, meaning the allocation of points, is performed by experts in expert interviews. Overall, seven interviews are conducted with experts, including two representatives of a regional energy utility company, one representative of an energy distribution network company, one management consultancy executive with a specialization in the energy sector, one representative of an NGO with an environmental focus, one member of the energy department of the German consumer protection organization and one representative of the German Trade Union Confederation. Based on the weightings defined in the interviews, several weighting profiles representing different perspectives are defined. In the context of this work, four profiles are defined, namely, energy utility, consumer (prosumer), environment NGO, and national economy. The differences on the category level are depicted in Figure 3.

These profiles are calculated by averaging the weightings over the relevant interview results. The energy utility weighting profile is based on the results of the two interviews with regional energy utilities and the distribution network operator. The consumer (prosumer) perspective is based on the interview results of the representatives of the consumer protection organization and the trade union. The environmental perspective is based on the interview with the environmental NGO. Finally, the national economy perspective is calculated as an average of all interview results with two exceptions. Firstly, the weighting results of the two participating regional utilities are averaged and counted as one result so that the utility perspective is not overrepresented in the national economy view. Secondly, the result of the consultancy executive is counted twice, as he has the broadest overview of

the energy sector. It is important to notice that results based on seven interviews cannot be statistically representative and, therefore, the weighting profiles need to be understood as a subjective but well-educated estimate.

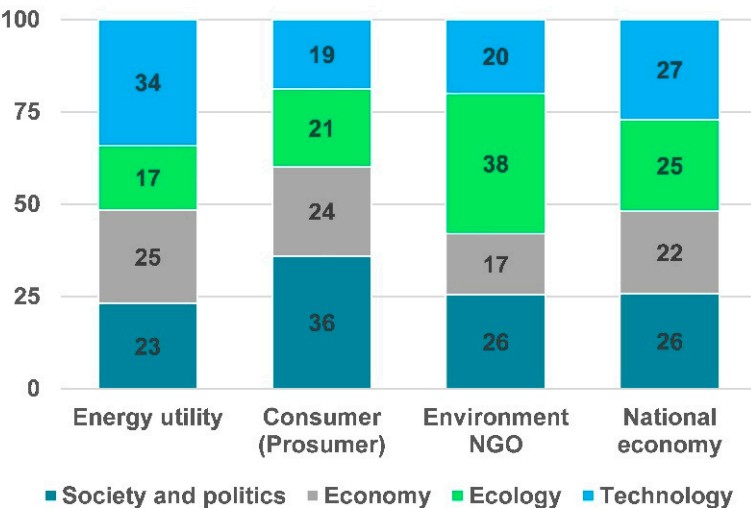

**Figure 3.** Weighting profiles.

### 2.4. Normalization and Aggregation of Results

Using the weights previously defined, the results can be aggregated. The aggregation is performed sequentially from sub-criteria to criteria, to category, to overall application level. Here, the simple additive weighting (SAW) method is used as described in [57] due to its simplicity. The aggregated result is calculated as the weighted sum of the (sub)criteria results by the following:

$$\overline{x} = \frac{\sum_{i=1}^{n} w_i x_i}{\sum_{i-1}^{n} w_i} \tag{1}$$

with $\overline{x}$ being the arithmetic mean, i.e., the result calculated based on the weight $w_i$ and the (sub)criteria evaluation $x_i$ with $i$ being the number of (sub)criteria.

In the case of the quantitative LCA results, these need to be translated into the qualitative MCA evaluation range before aggregation, referred to as normalization. Normalization is conducted using a reference value. The reference can be external (e.g., national $CO_2$ reduction target) or internal ($CO_2$ eq. reduction of a presented scenario). External referencing is preferred to internal references, as it offers increased objectivity [67]. However, the reference needs to be defined for each evaluation and each criterion based on the availability of information and objectives of the evaluation, external references may not always be available.

## 3. Test of the Proposed Framework

After the development of the evaluation framework and its adaption to digital applications in the energy sector described in Section 0, a digital application is evaluated as a first test of the framework's suitability. The evaluated digital application is the smart meter roll-out in Germany. This application is well defined by a German federal law [68]. As it is one of the main topics of the digitalization of the energy sector, it offers reasonably good availability of data.

### 3.1. Analysis of Boundaries and Assumptions

In the following section, the smart meter roll-out, the analysis boundaries, and key assumptions are described.

The subject of the evaluation is the smart meter in comparison to the conventional meter. The comparison is conducted in the context of the German national smart-meter

roll-out over a time period of 20 years. Operational roll-out effects (e.g., replacing still-functional conventional meters, system effects which require a minimum number of smart meters to become effective) are not considered, but rather a steady-state of using smart meters vs. using conventional meters.

The basic functionalities of the smart meter and its components (§21) as well as the smart meter roll-out (§29) in Germany are defined by the federal law for "digitalization of the energy transition" [68]. According to §21, a smart meter needs to have the functionality to measure the electricity used or produced and relevant grid parameters to store and manage the data, to provide an option to visualize energy-related data, to establish secure connections to transmit energy-related data, to connect further devices and to provide an option to implement different types of tariffs. The smart meter herein is called the intelligent measuring system (iMSys) and consists of the modern measuring device (mMe) and the gateway (GW), which is the communication unit. At this point in time (26 November 2020), there are four producers of smart meters offer devices, which meet the above criteria and are certified as defined by the German federal office for information security, while five others are still awaiting certification. The smart meter configuration modeled in this study is based on the information provided by two manufacturers exclusively to this research project.

The list of assumptions on aspects of the smart and conventional meters, such as the electricity consumption per device, lifetime, and the recycling share, is displayed in Table A1 (Appendix B). As no reliable information could be obtained on the maintenance need, the maintenance processes are not included in the LCA model. Several assumptions are taken regarding the as-is and future development of key aspects of the energy sector, including the number of metering points, the electricity demand, and the electricity GHG intensity.

The above-mentioned federal law also describes the roll-out plan. Energy consumers of more than 6000 kWh/year are required to receive an iMSys, while all smaller consumers receive only the mMe without the communication device. Energy producers with a bigger capacity than 7 kW are also required to install an iMSys. However, a revision of the German law for renewable energies [69], which is currently debated, envisions a roll-out of iMSys to generation assets of above 1 kW. The present study assumes a roll-out as described in the "Rollout Scenario Plus" of [19]. Based on this scenario, there will be installed 16.2 Mio iMSys and 35.4 Mio mMe, thus a total of 51.6 million meters in Germany. One difference between the used scenario and the currently planned roll-out exists. In the "Rollout Scenario Plus", it is assumed that all generation assets with >0.2 kW receive an iMSys instead of the currently planned limit of >1 kW. However, the difference in terms of number of smart meters as well as in terms of system impact is believed to be small enough to be disregarded. According to data of the German registry for electricity generators available at [70] in 2020, there are only ~29 thousand generation assets that fall into the described range versus a total of 1.52 million assets under the definition of [69].

The electricity demand in Germany is modeled based on the "Energy market prognosis" [71] of the German Federal Ministry for Economic Affairs and Energy. On the one hand, efficiency increases have reduced the demand in recent years and further efficiency increases can be expected during the next decades. On the other hand, a strong electrification trend is expected, e.g., electric cars, electric heating, which would increase electricity demand. Therefore, overall demand is expected to remain constant over the next 20 years.

The development of the GHG intensity of the German electricity mix is estimated based on historic emission and energy consumption data from the European GHG Inventory [72] and yearly report of the specific $CO_2$ emissions of the German energy mix published by the German Environment Agency [73] as well as the reference prognosis and trend scenario for future energy generation in [71] adopted for the 2019 developed coal exit path described in the final report of the German "Coal Commission" [74]. In contrast to the referenced climate reports, in this LCA, the GHG intensity is calculated as the global warming potential over 100 years (GWP100—calculated according to the "AR5"

dataset given in [75]). The GWP100 also includes effects of upstream value stream steps, e.g., the proportionate use of the electricity grid. Thus, the GHG intensity values calculated based on the GWP100 are higher than those ones of the referenced data. The GWP100 is measured in kg $CO_2$ eq. emissions (kg $CO_2$ eq.), which means that also GHG emissions other than $CO_2$, e.g., methane, are included, normalized to the effect of $CO_2$. Hence, in the following, if $CO_2$ eq. emissions are mentioned, the underlying calculation is the GWP100. This approach leads to an GWP100 of the German electricity mix of 0.460 (2020), 0.330 (2030) and 0.257 kg $CO_2$ eq./kWh (2040) as stated in Table A1. For all production process steps, the energy mix of 2020 is used; for the use-phase steps, the energy mix of 2030 is used; and for the end-of-life phase, the energy mix of 2040 is used.

In order to include the impacts of the smart meter on the energy system, a consequential LCA is performed. The question of how smart meters impact the energy system is discussed widely and several studies have come to different conclusions. In this study, two technical impacts on the energy system are considered: a reduction of the energy demand and a reduction of the required grid reinforcements. These are described as the most relevant technical impacts in the German smart meter cost–benefit analysis [19]. The reduction in energy demand is believed to be due to increased transparency on energy consumption of different appliances, leading to a behavioral change. The magnitude of behavioral change seems to depend on the level of feedback that a user gets from the smart meter, i.e., feedback via in-house displays or online platforms accessible via mobile devices result in more energy savings compared to feedback via the integrated display of the meter, which is usually located in a non-living area of the building. A good overview of different evaluations of the expected energy savings is given in ([20], p. 19 Part II). The authors cite 57 different studies indicating energy savings between 0 and 19.5%, with the most reliable studies ranging between 1 and 4%. The authors conclude to use 2.8% for the British cost–benefit analysis. For Germany, the authors of [19] conclude to take a more conservative approach due to the high level of uncertainty and assume an average energy saving of 1.8%. This study follows this overall conclusion. However, several aspects that may influence the overall energy savings are not modeled separately, such as different types of user feedback, the difference in savings between mMes and iMSys and the size (in kWh) and location of the smart meter user. Since the uncertainty of this assumption is high, especially because it is unclear whether the energy savings are time persistent, a sensitivity analysis with a focus on the energy savings assumption is conducted.

Besides the potential energy savings, the smart meters could also reduce the necessity for grid reinforcements. Grid reinforcements could be avoided due to improved planning based on more data points from the smart meters, due to reduced peak loads based on remotely controllable, small-scale distributed generation assets, and due to the reduction in energy demand. In [19], the authors quantify the reduction based on these three effects as 1% in the high voltage transmission grid and 30% in the low and medium voltage distribution grid. The reduction potential in the distribution grid is much greater, as the controllable small-scale generation assets are usually connected to the low voltage grid. This study follows the conclusion on grid reinforcement reduction taken by the authors of [19].

Although a variety of other impacts are likely to materialize, they are not included in the LCA as they are difficult to quantify, and no consistent values are found in the literature. Among these impacts is the increased integration of renewable energies, which could lead to a $CO_2$ eq. emission reduction and the effects of improved load shifting, which could further increase the reduction of the required grid reinforcements. Instead, these effects are reflected in the qualitative expert interview evaluations.

### 3.2. Life Cycle Assessment

A consequential life-cycle assessment is performed for the smart meter roll-out. For this purpose, first, the life cycle of one individual smart meter and one conventional meter (as a comparison) is analyzed. In a second step, these findings are put into the context of

the German electricity system over an analysis period of 20 years, which roughly resembles the lifetime of electricity meters. In a last and final step, the effects (consequences) that the smart meters cause in the energy system regarding energy demand and the need for grid reinforcements are included. A schematic overview is depicted in Figure 4.

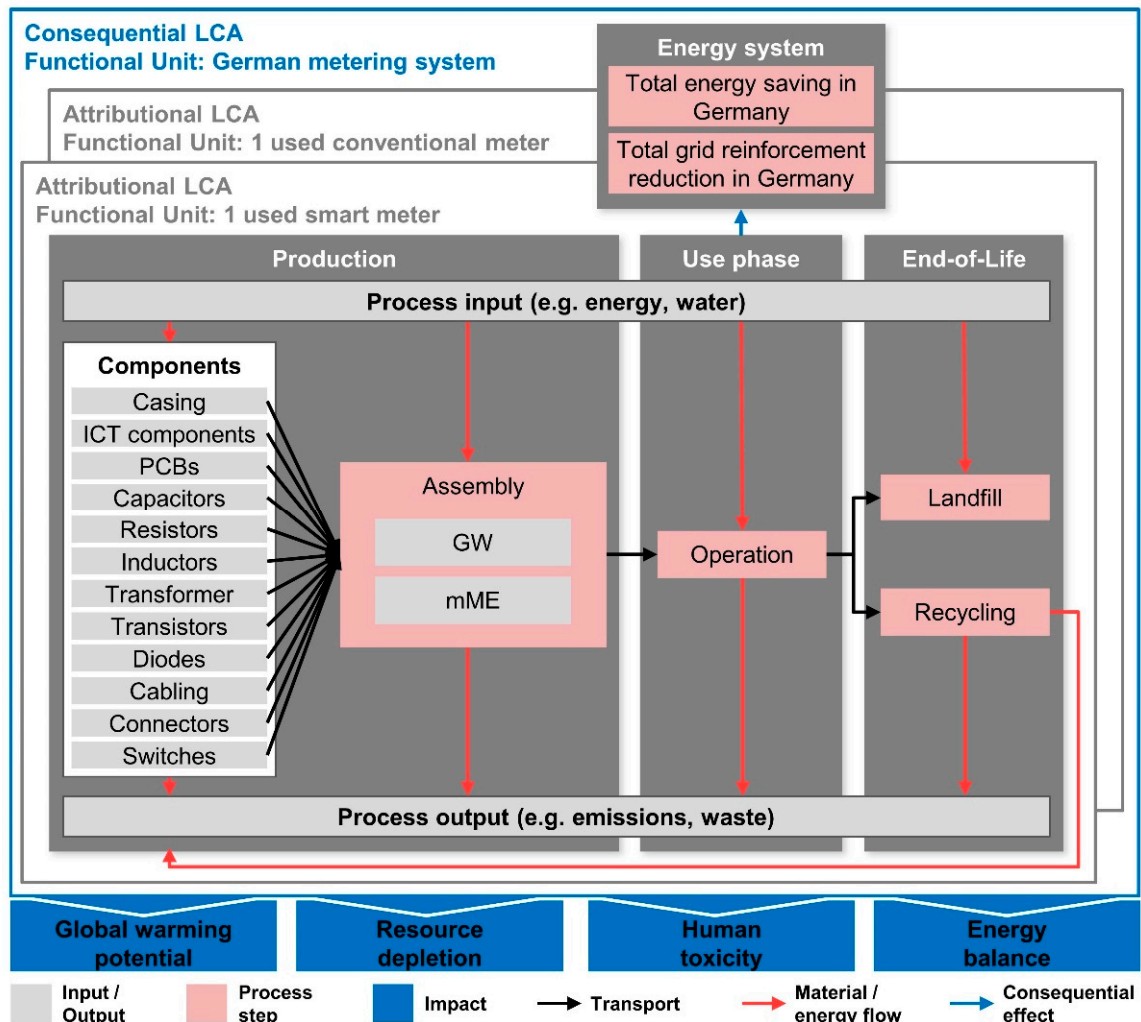

**Figure 4.** Schematic overview of smart meter LCA.

The life-cycle analysis of an individual smart meter (consisting of a mMe and a GW) is conducted in cooperation with two smart meter manufacturers. While one shared rather high-level results of the previous internal life-cycle analysis, the second one provided full transparency on the list of components assembled in the devices and their geographic origin (data supplied in several iterations between January 2019 and December 2020). All standard components, such as resistors, capacitors, and transistors (see Figure 4), are detailed based on the specific components' datasheets. All custom components, such as the device case, are detailed by the smart meter manufacturer.

The entire list of components and their origins is modeled in OpenLCA (1.10.1), using the ecoinvent (3.3-cut-off) database and a slightly modified version of the CML 2001 impact assessment method. The standard settings for production of the identified components are mostly used; however, the geographic location is adapted and the required transport is added. For some components, no standard settings are available; here, new settings based on literature and web research about the components' production process are implemented.

The main impact category discussed in the following section is the global warming potential over 100 years (GWP100). Further analyzed parameters are the energy balance,

human toxicity (HTP100), and the depletion of resources as the most relevant expected ecological impacts caused by digital applications in the energy sector.

The life cycle analysis results reveal that 75% of the GWP100 of the smart meter's production is caused by the gateway, with the biggest contributor for both devices, GW and mMe being the ICT components. The total production-related GWP of one smart meter is found to be 82 kg $CO_2$ eq. When the use phase as well as the recycling/landfilling are added to the LCA, the total smart meter lifetime GWP is calculated to be 558 kg $CO_2$ eq. Similarly, to the smart meter, the conventional meter is modeled. However here, data of incorporated materials from [43] are used for the life-cycle assessment. The overall production GHG footprint of a conventional meter is found to be 11 kg $CO_2$ eq., and therefore much smaller than the footprint of the smart meter. The total conventional meter lifetime GWP is 211 kg $CO_2$ eq.

In order to directly compare the smart meter and the conventional meter, their lifetime needs to be proportionally considered, i.e., 20/18 = 1.11 smart meters and 20/30 = 0.67 conventional meters are required during the 20-year analysis period. The result shows that the use of a smart meter causes an increased GWP100 impact of about 489 kg $CO_2$ eq., more than the use of a conventional meter over 20 years. However, not all consumers will receive fully-fledged smart meters; some, depending on their consumption, will only receive a mMe without the GW. Therefore, the roll-out assumption about the number of mMes and full smart meters (iMSys) described in Section 0 need to be considered. Approximately 35.4 million mMes and 16.2 million smart meters (iMSys), instead of a total of 51.6 million conventional meters, are assumed to be in use during the analysis period of 2020–2040. The life cycle impact (including production, use-phase, and end-of-life) of the smart meter roll-out, compared to using conventional meters, is an increase in the GWP100 of $8.2 \times 10^9$ kg $CO_2$ eq., i.e., 8.2 Mio t $CO_2$ eq. over 20 years.

However, the use of smart meters also causes effects within the energy system, which need to be included in the consequential LCA. As discussed in Section 0, the two system effects of electricity savings and reduction of grid reinforcements are modeled. Taking into account the life cycle impact of smart and conventional meters and the positive system effects of smart meters, overall, a GWP100 reduction of 23.9 Mio t $CO_2$ eq. over the analysis period of 20 years is identified.

The GWP100 results of the LCA of the live cycle comparison between smart and conventional meters are depicted in Figure 5. GWP100 impacts of the conventional meter are regarded as negative, as they are seen as avoided impacts.

Besides the GWP100, two further LCA impact categories, human toxicity, and resource depletion, as well as the LCA energy balance, are analyzed. Under the above-described assumptions (base scenario), all parameters show a significantly positive impact of the smart meter roll-out (see Figure 6).

The sensitivity of the results to different assumptions is discussed in Section 3.4 (see Figures 9 and 10).

### 3.3. Expert Interviews and MCA Evaluation

The expert interviews to evaluate the smart meter roll-out in Germany are conducted with the same seven experts already consulted to develop the weighting profiles (see Section 0). In some cases, weighting and evaluation are performed within one session. For every criterion, the experts could decide whether they want to provide any evaluation at all and, if so, if they want to evaluate on the criteria level or, in case they feel sufficiently knowledgeable, on the sub-criteria level. This approach ensures that experts can evaluate criteria in-depth in their area of expertise, while avoiding evaluating criteria where they lack the required expertise.

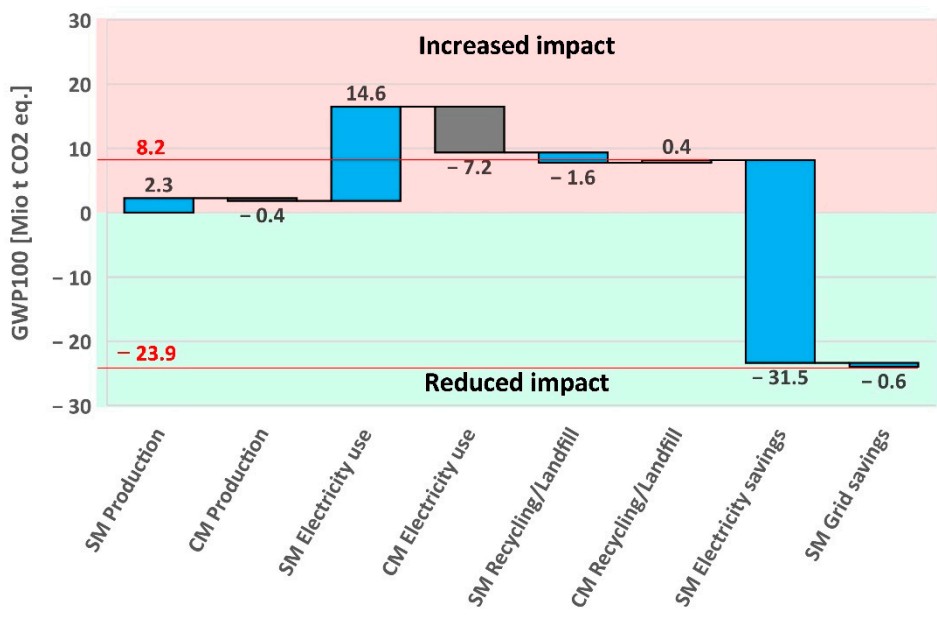

**Figure 5.** GWP100 result of LCA comparison of smart and conventional meters over 20 years.

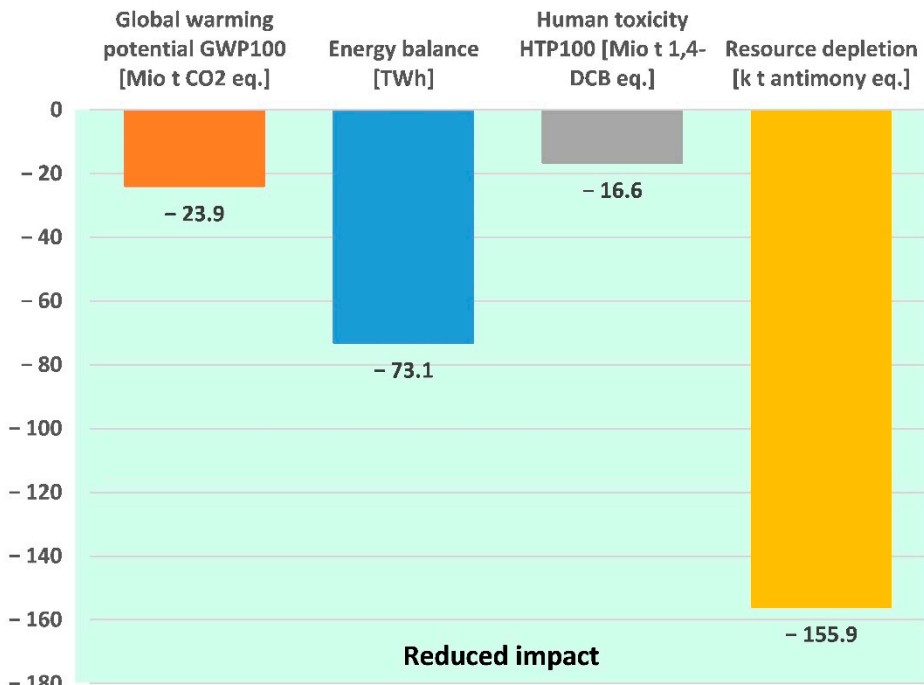

**Figure 6.** LCA impact categories for comparison of smart and conventional meters over 20 years.

For the aggregation of results, the SAW method is applied (see Section 2.4), using the weights of the defined weighting profiles (see Figure 3).

As one of the goals of the presented evaluation framework is to combine the views of different stakeholders, the overall evaluation is calculated based on the arithmetic average of each sub-criterion, criterion, and category across all experts. One adaption, however, is made. Since two of the experts are representatives of a regional energy utility, their results are averaged first such that they only contribute as one expert to the overall result (same procedure as conducted for calculating the weighting average).

For LCA criteria, i.e., energy balance, GWP100, resource depletion, and human toxicity, only the normalized LCA results are used for the MCA evaluation. The quantitative LCA results need to be translated into the qualitative MCA evaluation scale from −3 to +3. As described in Section 2.4, the reference for this normalization needs to be determined for each criterion and for each assessment of a digital application. In this case, the results are translated against an internal value, the results of the "2.25% electricity savings" scenario, which reflects a very positive scenario. The scale is set to be +3 for the results of the "2.25% electricity savings" scenario and 0 for zero effect on the impact category. For the base scenario, this translates into MCA evaluations of +2.3 for the energy balance, the GWP100, and the resource depletion and a +2.5 for human toxicity.

Overall, the above approach leads to the results depicted in Figures 7 and 8. While Figure 8 provides a better overview of aggregated results, Figure 7 is particularly important to understand which benefits are generated by the application and where risks of negative impacts appear. This knowledge can start a solution-finding process to resolve the causes of negative evaluations.

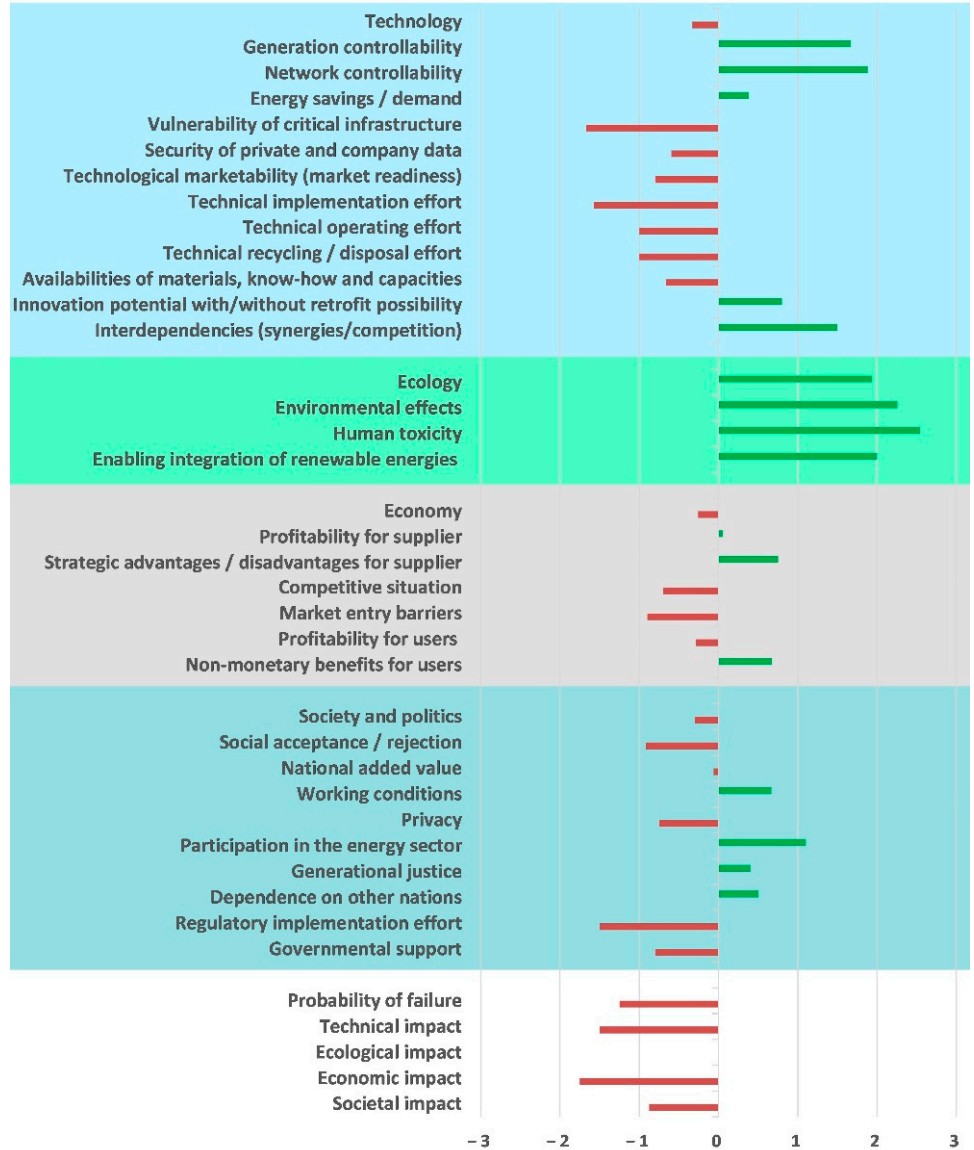

**Figure 7.** MCA result per criterion for national economy weighting profile.

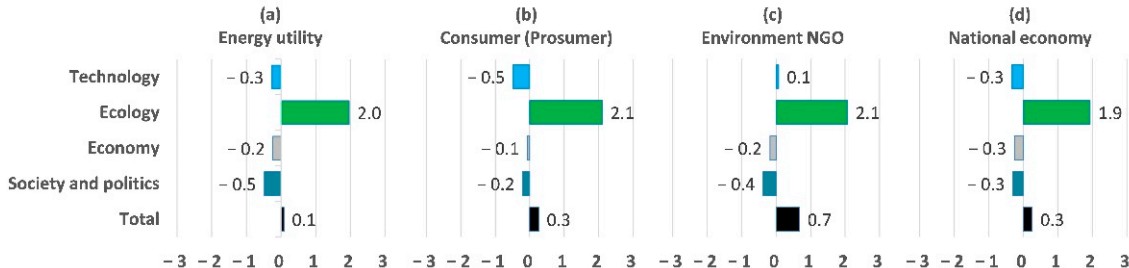

**Figure 8.** MCA result per category for four weighting profiles.

The results per criterion for the national economy weighting profile are depicted in Figure 7. The categories technology, economy, and society and politics show an overall negative evaluation while ecology overall is evaluated positively.

In particular, the category technology reveals a negative result. The main negatively evaluated criteria are the impact on the vulnerability of the critical infrastructure and the technical implementation effort. On the side of positively evaluated criteria, the effects on the controllability of the generation and the grid are among the most positive. Further positively evaluated criteria are the innovation potential and interdependencies.

The ecology result is significantly positive across all criteria and sub-criteria.

The economy category is evaluated as slightly negative, mainly due to high market entry barriers and expected future increased competition. The most positive criteria of this category are the non-monetary benefits for the supplier and the user. For the supplier, the potential to reach new customer segments and for the user an increase in convenience are the most positive non-monetary benefits. The monetary profitability for the supplier is very slightly positive. On the other side, the profitability for the user is negative.

In the category of society and politics, the experts see a significant need for regulatory implementation effort and the need for governmental support. Furthermore, it is expected that the smart meters will be met with rejection rather than with acceptance. The main positive criterion of this category is the increase in participation of formerly passive consumers in the energy sector.

For the evaluation of the probability and impact of potential failures, the worst technical malfunction is considered. For the smart meter roll-out, this worst case is assumed to be a widespread malfunction of the meters, delivering no or wrong consumption, generation, and grid data. The case of an intentional attack on smart meters is already covered in the criterion vulnerability of the critical infrastructure. The biggest negative impacts are believed to be of an economic and technical nature, while the societal impact is only slightly negative, and no ecological impact is expected. As risk is a cross-functional category, no category score is calculated, but the evaluations are allocated to the respective category where the impact occurs.

As discussed in Section 2.3, different weighting profiles are derived from the weighting expert interviews (see Figure 3). These weighting profiles influence the overall MCA results, as they represent the perspectives of different stakeholders. The different results are depicted in Figure 8. Figure 8d, "National economy", shows the same weighting profile used for the results in Figure 7. The energy utility perspective (a) is notably similar to the national economy perspective. The main difference is in the evaluation of the category society and politics. Here, the energy utilities give more weight to the regulatory implementation effort, which causes a more negative overall result. The consumer perspective (b) reveals a more negative result in the technology category. This is mainly due to the higher weighting of the IT and data security criteria. The environmental NGO perspective (c) is the only one with a positive result in the technology category, due to the much lower weighting of the technological effort criteria (e.g., implementation effort, operation, and maintenance effort); this gives more relative weight to the positively evaluated generation and grid controllability. The positive technology evaluation and more weight on the ecological category cause the more positive evaluation of this weighting profile.

Overall, the total result is slightly positive for all weighting profiles. The environmental impact category is evaluated positively for all weighting profiles, while the economy and society and politics categories are evaluated negatively for all weighting profiles. The technology category is only positive when evaluated with the environmental NGO weighting profile.

The different weighting profiles could be seen as a sensitivity analysis regarding the impact of weighting. However, in the proposed framework, the use of different weighting profiles is an integral part, thus these weighting profiles and their effects are seen as part of the results rather than a sensitivity analysis.

### 3.4. Sensitivity Analysis

First, the sensitivity of the LCA results to two central assumptions (potential contribution to energy savings, decarbonization paths of the German energy mix) is analyzed, and subsequently, the impact of different LCA results on the overall MCA evaluation is assessed.

Although the LCA is based on a detailed technical analysis of the components of a mMe and a GW, several assumptions need to be made when the findings of these devices are put into the context of a nationwide roll-out with effects on the broader energy system. The main assumptions are discussed in Section 3.1. An overview of all explicit assumptions is presented in Table A1.

Figure 9 depicts the sensitivity of the LCA result regarding the assumption on the smart meters' potential to cause energy savings. The reasons why a special focus is put on this assumption are that, on the one hand, it is subject to high uncertainty, and, on the other hand, it has a very strong effect on the LCA outcome. The base scenario assumes a saving of 1.8% of the electricity used by the users of the smart meters. In this scenario, all evaluated LCA impact categories are positively impacted (meaning a reduction of the impact is achieved). Therefore, the sensitivity analysis focuses primarily on identifying the tipping point where LCA results turn out negative (meaning an increase of the impact). Within the analyzed range of energy savings, all four impact factors have a linear correlation with the energy-saving assumption. The GWP100 break-even energy saving rate is found to be 0.43%. If smart meters generate more energy savings, their overall impact on the GWP100 is positive (GWP reduction). The energy balance and the resource depletion break-even point is very close to the GWP100 break-even point, indicating the close connection of the impact factors. Regarding the impact on human toxicity, a decrease is identified across all energy saving scenarios, even for the no electricity savings scenario. This is due to the modeled grid reinforcement reduction, which reduces the use of copper and its effects on human toxicity.

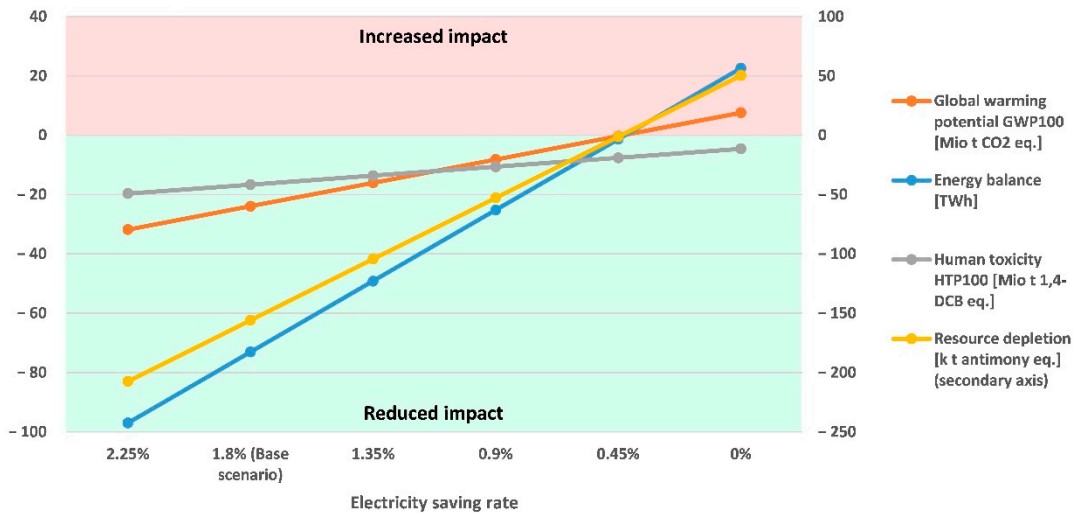

**Figure 9.** Sensitivity of LCA result regarding energy saving assumption.

Since the electricity savings determine whether an overall GWP reduction can be achieved, an analysis of the different decarbonization paths of the German energy mix is necessary. If the decarbonization is achieved faster than expected, i.e., GWP intensity is only half of the estimates for 2030 and 2040, a GWP reduction would still be achieved. However, with only 12.02 Mio t $CO_2$ eq. over 20 years compared to 23.9 Mio t $CO_2$ eq. in the base scenario, the reduction is significantly smaller. If the GWP intensity does not decrease but rather stays constant over the next 20 years, the potential emission reduction increases to 33.5 Mio t $CO_2$ eq., respectively.

Comparing the influence of the energy saving on the LCA outcome in Figure 9 to the further impact factors in Figure 10, it can be seen that the very dominant effect on the impact categories is created by the energy savings. The "no grid reinforcement savings" scenario slightly reduces the savings in resource depletion due to the use of copper wire for grid reinforcements. This, in turn, also reduces the positive impact on human toxicity. However, the GWP saving remains almost unchanged. The "no recycling of smart meters" scenario tests the impact if smart meters cannot be recycled. Overall, the impact of this scenario is rather low, mainly due to the low production impact and long lifetime of the devices. If the smart meters have an own electricity consumption twice as high as assumed, a stronger effect on the energy balance, the resource depletion, and the GWP can be noted. An even higher own electricity demand is very unlikely, according to the manufacturers and utility experts. Yet, this effect does not come close to eliminating the overall savings. In the case of no positive system effect, i.e., no electricity savings and no grid reinforcement savings, all LCA impact categories increase and consequently, no positive impact is generated by the smart meters. The negative impact seen in this scenario is the production and operation impact of smart meters as electronic devices with a higher own electricity consumption and a shorter lifetime, compared to the analog conventional meters.

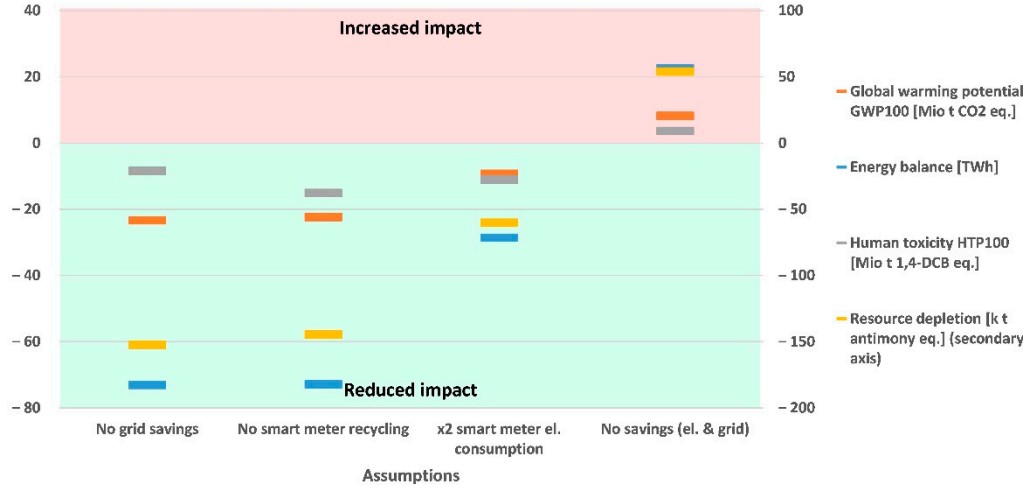

**Figure 10.** Influence of general assumptions on LCA result.

In a second step, the effect of the LCA result on the overall MCA evaluation is assessed. As described in Section 0, the LCA results are translated against an internal value, the results of the "2.25% electricity savings" scenario. While for the base scenario, this translates into MCA evaluations of +2.3 for the energy balance, the GWP100, and the resource depletion and a +2.5 for human toxicity, for the worst-case "no savings" scenario (no electricity savings and no reduction of grid reinforcements) this translates into an MCA evaluation of −0.7 for energy balance, −0.8 for GWP100 and resource depletion and −0.6 for human toxicity. Based on this translation, the difference between the base scenario and the "no savings" scenario affects the MCA result regarding the ecology category, which changes from an overall score of positive 1.9 to a neutral 0 for the national economy perspective. This, in turn, changes the overall MCA outcome from a positive 0.26 to a negative −0.23.

## 4. Discussion

Following the structure of the former section, the discussion is structured in three parts: LCA, MCA, and sensitivity analysis.

### 4.1. LCA

Smart meters are relatively similar to other ICT hardware with regards to the type of components and materials used and the manufacturing process; therefore, the total production GWP impact and the main contributors are similar. A smartphone has a production GWP of approximately 48 kg $CO_2$ eq. [76]. Considering that a smartphone is significantly smaller and lighter but has a more potent CPU and a bigger display, the obtained smart meter GWP result of 82 kg $CO_2$ eq. is plausible. Besides that, similar to the smartphone, the main source of GWP impact is identified as the ICT components, such as integrated circuits. In contrast to a smartphone, the electricity meters have a long lifetime; therefore, the share of the GWP impact of the production, compared to the use phase, is found to be rather small. With an expected lifetime of 18 years, the GWP100 of the smart meter use phase is by a factor of seven greater than the GWP100 of the production of the device. This finding is roughly confirmed by the high-level life cycle analysis data supplied to the research project by a second smart meter manufacturer. Thus, own electricity consumption is the factor with the biggest impact on the GHG emissions caused by the electricity meters and, therefore, offers the most significant leverage for improvements.

Comparing the life-cycle impact of using smart meters over 20 years to using conventional meters reveals an increase in the GWP100 of 8.2 Mio t $CO_2$ eq. This shows that if the smart meters do not generate benefits, they merely increase the GWP100 of the metering system. However, the additional 8.2 Mio t $CO_2$ eq. emissions over 20 years are small compared to the yearly ~111 Mio t $CO_2$ eq. emissions (calculated for 2020 based on assumptions in Table A1, 104 Mio t $CO_2$ eq. based on [71]) of the electricity consumed by the smart meter users (households and small businesses). Therefore, it is apparent that already, a very small impact on the energy demand can neutralize the additional GWP100 emissions of the smart meters.

The consequential LCA looks beyond the direct life-cycle impact of the device and takes system impacts into consideration. In this case, the two system impacts energy savings and grid reinforcement reductions are assessed. When including both of them, the overall effect of using smart meters instead of conventional meters is positive, meaning a GWP reduction of 23.9 Mio t $CO_2$ over 20 years. Here, the overwhelming effect is the reduction of the energy demand. However, comparing the GHG reduction achieved over 20 years to the yearly $CO_2$ eq. emissions of the smart meter users (111 Mio t $CO_2$ eq., see above) reveals that the smart meter, including the modeled system effects, does not drastically change the $CO_2$ eq. intensity of the energy mix. In order to play an important role in the energy transition, the smart meters need to enable further system effects, such as enabling the integration of more renewable energies.

Besides the GWP, three additional impacts are assessed. Under the base scenario assumptions, all parameters show a significantly positive impact of the smart meter roll-out. The reduction of 73.1 TWh of the electricity demand is driven by the modeled energy savings of 1.8%. The reduced human toxicity is due to the modeled energy savings (~2/3) and the modeled grid reinforcement reduction (~1/3). The reduced resource depletion is mainly driven by the reduced energy demand and the subsequently reduced depletion of fossil fuels. The different hardware of smart and conventional meters does not have a significant impact on the overall resource depletion.

Although the results of the LCA of smart meters vs. conventional meters in California [43] partly include different effects and therefore indicate different GWP levels, qualitatively, they are in line with the results of this study, e.g., proportions of GWP contribution of production, use-phase, and end-of-life, offset of higher GWP of smart meters production and use-phase via energy savings.

*4.2. MCA*

The MCA discussion is structured along the four criteria categories. The results are based on the expert interviews as well as the LCA results.

The category technology is evaluated negatively, thus it is necessary to identify which aspects cause this negative evaluation and identify potential solutions. The results show that the experts mainly see a high technical implementation effort and a negative effect on the vulnerability of the critical infrastructure. These points indicate areas where a special focus needs to be applied when defining the smart meter roll-out. Furthermore, some steps are seen as still necessary to reach marketability of the smart meters. This includes the technical ability to allow remote control, the interoperability between devices of different manufacturers and a process to recalibrate existing meters. On the other hand, also positive impacts can be identified. The main positively evaluated technological criteria are the controllability of the electricity generation and the grid. However, experts consistently noted that one requirement for these positive impacts to materialize is an extended possibility for distributed generation assets and consumers to be remotely controlled. This involves establishing the regulative basis (currently not given for consumers and smaller generators) and the certification of communication gateways with the required technical ability. On the one hand, the remote controllability is the basis for several other intended positive impacts of the smart meter, such as the integration of renewable energies, energy savings, improved profitability for users due to flexible tariffs and, in a broader sense, increased participation of users in the energy sector. On the other hand, however, the requirement to equip also smaller generation assets with smart meter gateways could reduce their profitability and cause fewer small-scale renewable assets to be connected to the grid. These effects on grid and generation controllability are the main technological reason for the smart meter implementation and as such, their influence on the evaluation of the technology category might be underrepresented. Experts consistently were surprised by the negative (or only very slightly positive) outcome of the technology category based on their own weighting and evaluation. A higher number of effort criteria, which are generally evaluated negatively, potentially causes this apparent weighting imbalance. Thus, a better balance between effort and impact criteria might be needed. Many experts see the smart meter as an enabler for future developments, such as smart home applications. Therefore, the innovation potential is evaluated positively. Nevertheless, the experts expressed the concern that the first-generation smart meters already need to be upgradeable/updateable to fulfill the full range of future applications and thus do not need to be replaced before the end of their lifetime.

Due to the positive LCA results and the positive effect on the integration of renewable energies, the overall ecological result is positive. The improved integration of renewables likely correlates with the technical criteria of generation and grid controllability and is one of the main reasons for the implementation of smart meters. The expert interview results for LCA criteria, although not part of the evaluation, can be used to crosscheck the LCA results. The experts evaluate the LCA criteria as less positive, compared to the outcome of the LCA. The biggest difference occurs for the sub-criterion resource depletion. Here, it is possible that the experts considered a practical roll-out, which involves replacing still-functional conventional meters with new smart meters, which influences resource depletion negatively. In the LCA, this practical roll-out effect is not considered.

The monetary profitability for the supplier is slightly positive. Considering that the prices for smart meters are regulated and set as low as possible while still allowing the supplier to benefit, this evaluation is plausible. The profitability for users, on the other hand, is evaluated slightly negatively. The experts do not expect that the users' reduction in the electricity bill (or increase in revenues from a small generation asset) will outweigh the monthly cost of the smart meter. However, many experts note that the profitability as well as the users' non-monetary benefits could be improved/increased if further applications are built on the basis of the smart meter, such as smart home systems. A key requirement for this is the right to use the user data, potentially by third parties, to offer these products

and services. For consumers with a small consumption who only have the mMe without the communication gateway, it is required to establish a connection to the mMe via, for example, the user's own router. Here, the regulatory framework needs to be established. However, as these measures could affect the users' privacy and the integrity of the user data, caution is required, and a compromise needs to be made. The negative profitability for the user in combination with concerns about data security and privacy could be the reasons for the experts' opinion that the smart meters will rather be rejected than accepted. Here, specific research into the reasons why users may reject the smart meter are recommended in order to develop measures to increase the acceptance.

Besides the results of the national economy perspective discussed in depth above, three additional stakeholder perspectives are defined as weighting profiles. In this case, all weighting profiles yield the same overall result tendency. However, the environmental NGO perspective reveals two differences. The overall result is more positive compared to the other weighting profiles and the technological category is positive instead of negative. This indicates that the biggest advantages are in areas important for these stakeholders, such as grid and generation controllability, integration of renewables, and life cycle performance. The most negative result in the technology category occurs in the consumer perspective (b) due to a higher weighting of the IT and data security criteria. This indicates that this is a point of concern for users and, as such, needs to be addressed by energy utilities as well as the regulator.

### 4.3. Sensitivity Analysis

In the sensitivity analysis, the energy saving parameter is identified as the most influential on the LCA result. The GWP100 break-even energy saving of 0.43%, above which an overall GWP reduction is achieved by the application of smart meters, is roughly confirmed by [43], where it is found to be 0.25%. Within the scope and range of the sensitivity analysis, no other factor changes the tendency of the result. In a second step, the influence of the LCA sensitivity on the MCA outcome is analyzed. In the case of the "no savings" scenario, both the ecology category and the overall result turn out negative. This result underlines the importance of ensuring that these benefits are, in fact, achieved during the roll-out.

### 5. Conclusions and Outlook

A framework consisting of an MCA combined with an LCA and expert interviews for the holistic evaluation of digital applications in the energy sector is proposed and tested on the use case of the smart meter roll-out. The framework provides a result, which is reduced in complexity to enable broader discussions, reflects the perspectives of different stakeholders and provides detailed insights on critical aspects of the application.

Criteria for the MCA are developed based on the review of literature and refined in expert interviews. Weighting profiles for perspectives of different stakeholders are developed, also based on expert interviews. The LCA is performed using detailed information on the technical setup of smart and conventional meters and a set of assumptions. The outcome of the LCA is the assessment of several criteria in the ecology category. LCA results are subsequently integrated into the MCA evaluation. All MCA criteria are evaluated in expert interviews. Results are aggregated using the SAW method.

The evaluation of the smart meter roll-out is the first test of the proposed evaluation framework.

The LCA reveals that the smart meter as an electronic device has a higher environmental impact compared to conventional meters, mainly due to the high consumption of electricity, which is the biggest impact factor on the smart meter's life cycle greenhouse gas emissions. However, if the smart meter roll-out results in an electricity saving of >0.43%, the overall environmental impact, mainly the GWP, would be reduced. The electricity saving potential has the most significant influence on the overall LCA result.

The overall MCA result, including evaluation of criteria in expert interviews and the LCA results, reveals a slightly positive overall score for most stakeholder perspectives. While the core reasons for the smart meter roll-out, generation and grid controllability, and further integration of renewable energies are all evaluated very positively, negative evaluations in the area of IT and data security, implementation and operational effort, social acceptance, and profitability overall offset the result into the negative range. These negative aspects need to be addressed and solved in order to take full advantage of the positive impacts of the smart meter. Critical aspects of the smart meter roll-out are identified, and key insights on required actions are gathered during the expert interviews.

Overall, the first test of the framework results in a largely consistent evaluation of the smart-meter roll-out, in line with the relevant discussed existing studies on smart meters for most aspects. Therefore, the overall goal of the proposed framework is met, and it is concluded that the framework is suitable for evaluating this digital application in the energy sector. Yet another test with a different application is recommended to prove the versatile applicability of the framework for digital applications.

Although the first test is considered successful, several areas for further improvements of the framework are identified. Most notably, some modifications need to be made on the criteria. It would be beneficial to reduce the number of overall criteria to improve the practical usability of the framework, although, due to the complexity and diversity of digital applications, no drastic reduction is possible. In the technology category, it seems to be necessary to rebalance the number of effort and impact criteria to obtain results that are more realistic. This could be done by summarizing several effort criteria into one. Furthermore, although the intention while defining the criteria was to avoid overlaps and, therefore, double counting as far as possible, some cases of double counting were identified and need to be resolved. The most prominent example is the technical impact on energy consumption and the ecological criteria energy balance. Lastly, the hierarchy level of some criteria and sub-criteria needs to be reconsidered, e.g., human toxicity and global warming potential should be on the same hierarchy level.

The point allocation weighting method based on the distribution of 100 points was easily understood. However, in practical use, many experts faced the difficulty of leveling the weights between (sub)criteria of different categories. To reduce this problem, some pairs of criteria were randomly picked for direct comparison. Sometimes the two criteria could meaningfully be compared; sometimes, the direct comparison did not seem suitable. It is possible to define a standard set of suitable (sub)criteria pairs for direct comparison, ensuring that general leveling of weightings is achieved.

The evaluation range set from −3 to +3 was easily understood and, for the most part, intuitively used. No need for adaption was seen here.

Currently, the results are presented on the criteria level and summarized to the category level, following the hierarchical structure of the criteria. However, it could be of significant interest to derive a medium layer of result aggregation, which does not follow the hierarchical structure but is based on cross-functional result categories. These result categories could summarize different efforts, such as implementation and operational effort as well as different impacts, such as effects on the system stability, the environment, or the IT security. Based on these result categories, a weighting sensitivity analysis could be performed (e.g., sensitivity if system stability is weighted more).

Currently, neither the uncertainty of the input nor the number of evaluation points (how many experts evaluated a specific criterion) is considered. Therefore, the robustness of the results can only be ensured through the sensitivity analysis. An indication of the robustness could make transparent where a sensitivity analysis is required and increase confidence in the result.

Furthermore, a check of the usage of materials versus a list of critical materials, such as [77], issued by the European commission could be performed. This is already partly covered in the LCA impact category depletion of resources. However, only a more in-depth knowledge of the exact material can lead to mitigation actions, such as finding substitutes.

Lastly, the weighting profiles should become adaptive not only to the stakeholder's perspective but also to the type of application under evaluation such that non-relevant criteria do not need to be evaluated.

**Author Contributions:** Conceptualization, P.W. and M.F.; methodology, P.W. and P.V.; software, P.W.; validation, P.W., M.F. and P.V.; formal analysis, P.W.; investigation, P.W.; resources, P.W.; data curation, P.W.; writing—original draft preparation, P.W.; writing—review and editing, P.W., M.F. and P.V.; visualization, P.W.; supervision, M.F. and P.V.; project administration, M.F.; funding acquisition, not applicable. All authors have read and agreed to the published version of the manuscript.

**Funding:** This research received no external funding.

**Data Availability Statement:** The source of all publicly available data used for the research is specified as a reference at the appropriate position within the paper. The information about the technical setup of the smart meter used in the LCA is obtained directly from a manufacturer under the condition of a non-disclosure agreement and cannot be published. Further information obtained by the manufacturer is listed in Appendix B.

**Conflicts of Interest:** The authors declare no conflict of interest.

## Appendix A

*Appendix A.1. Definitions of Digitalization Terminology*

The term digitalization as well as connected terms, such as digital transformation, digital applications, or the term "smart", have gained popularity during the last decade. Monthly online searches for the term digitalization have increased by a factor of 8 between 2010 and 2020 [78]. However, there is still no common definition of the key terms. Therefore, these are defined for the context of this study. The definition is based on the previous publication [5] and is very similarly used in [79].

*Appendix A.2. Digital Technology*

Digital technologies can be based on software or a combination of hardware and software, so-called cyber–physical systems. Pure ICT (Information and Communication Technology) hardware thus only becomes a digital technology in combination with the corresponding software. Software for which no specific hardware is required can be defined as a software-based digital technology, but of course, some kind of hardware is also required for its execution. Even given this definition, different authors see different digital technologies. Overviews of digital technologies are presented in [79] (which reflects the authors' view based on a survey among European utilities), a survey among German medium-size utilities ([80], p. 22), an extensive book on digital transformation ([81], p. 31 ff) and a global survey among companies of the energy sector ([82], p. 9).

*Appendix A.3. Digitalization*

The term digitalization includes both the increasing implementation of digital technology in more and more areas of business and private life as well as the resulting socio-economic effects in the economy as well as the society.

*Appendix A.4. Digitization*

Digitization is the process of changing from analog to digital, e.g., converting paper documents into pdf (or other formats) or creating digital data tables from paper-based data. Therefore, digitization can be seen as one of the initial steps of digitalization.

*Appendix A.5. Digital Transformation*

The digital transformation describes the socio-economic part of the process of implementing the digitalization within companies or other groups. In particular, this includes the creation of digital strategies, the adaptation of new working methods and structures, and, in a broader sense, cultural and organizational changes.

### Appendix A.6. Digital Application

Digital applications must be based on digital technologies and are usually the goal of digital transformations. The distinction between technology and application is made by the question of whether there is a direct benefit. An application must generate a benefit in itself, while a technology forms the basis for applications but does not have an inherent benefit. Of course, some applications and technologies can be argued to be either one.

### Appendix A.7. Smart

The term smart, although very frequently used, still lacks a common definition. For the context of this article, the authors assume the following characteristics for the adjective smart as defined in [5]: (1) being digital (in contrast to analog), (2) being connected via communication technology, and (3) being able to process information (locally or in the cloud).

### Appendix B

**Table A1.** List of LCA assumptions.

| | Unit | Production (Value in 2020) | Use-Phase (Value in 2030) | End of Life (Value in 2040) | Source |
|---|---|---|---|---|---|
| **Assumptions German electricity system** | | | | | |
| GWP100 German electricity mix p.a. | kg $CO_2$ eq./kWh | 0.460 | 0.330 | 0.257 | See Section 0 |
| Electricity use households & small businesses p.a. | TWh | 265.7 | constant | | [83] * |
| Grid reinforcement low voltage p.a. | km | 690 | constant | | [84] ** |
| Grid reinforcement medium voltage p.a. | km | 2920 | constant | | [84] |
| Grid reinforcement high voltage p.a. | km | 490 | constant | | [84] |
| Number of mMes | | 35,400,000 | constant | | [19] |
| Number of iMSys | | 16,200,000 | constant | | [19] |
| Number of conventional meters | | 51,600,000 | constant | | [19] |
| **Assumptions smart meter devices** | | | | | |
| Analysis period | years | 20 | - | | - |
| Lifetime of mMe | years | 18 | constant | | Grid operator |
| Lifetime of GW | years | 18 | constant | | Grid operator |
| Lifetime of conventional meter | years | 30 | constant | | Grid operator |
| Effective power mMe | W | 2.4 | constant | | Manufacturer |
| Effective power GW | W | 8 | constant | | Manufacturer |
| Effective power conventional meter | W | 2.4 | constant | | Grid operator |
| Energy use of production mMe | kWh | 1 | constant | | Manufacturer |
| Energy use of production GW | kWh | 1 | constant | | Manufacturer |
| Energy use of production conventional meter | kWh | 1.6 | constant | | [43] |
| Energy use of disassembly mMe | kWh | 3.3 | constant | | Ecoinvent |
| Energy use of disassembly GW | kWh | 2.2 | constant | | Ecoinvent |
| Energy use of disassembly conventional meter | kWh | 10.9 | constant | | Ecoinvent |
| Weight mMe | g | 495 | constant | | Manufacturer |
| Weight GW | g | 339 | constant | | Manufacturer |
| Weight conventional meter | g | 1650 | constant | | [43] |
| Recycling rate SM | % | 80% | constant | | Manufacturer |
| Recycling rate conventional meter | % | 80% | constant | | Estimate |

**Table A1.** *Cont.*

| | Unit | Production (Value in 2020) | Use-Phase (Value in 2030) | End of Life (Value in 2040) | Source |
|---|---|---|---|---|---|
| **Assumptions effects of smart meter** | | | | | |
| Electricity demand savings | % | 1.8% | constant | | [19] |
| Electricity demand savings p.a. | TWh | 4.782 | constant | | calculated |
| Grid reinforcement reduction low voltage | % | 30% | constant | | [19] |
| Grid reinforcement reduction medium voltage | % | 30% | constant | | [19] |
| Grid reinforcement reduction high voltage | % | 2% | constant | | [19] |
| Grid reinforcement reduction low voltage p.a. | km | 207 | constant | | calculated |
| Grid reinforcement reduction medium voltage p.a. | km | 876 | constant | | calculated |
| Grid reinforcement reduction high voltage p.a. | km | 9.8 | constant | | calculated |

\* Data on net electricity consumption by consumer groups provided by the German Association of Energy and Water Industries; \*\* study on the innovation requirements in the German electricity distribution network by the German Energy Agency.

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
