# Peer review of "Holistic Evaluation of Digital Applications in the Energy Sector—Evaluation Framework Development and Application to the Use Case Smart Meter Roll-Out"

_sustainability, doi:10.3390/su13126834_

Round 1
Reviewer 1 Report
I consider that this research paper is very interesting with very interesting tables and figures. Nevertheless, I think there are some improvements to be made, namely:
- The title can and should be improved
- It is necessary to complete the text with more bibliographical references
-Improvements need to be made in the way it presents the bibliography, particularly with regard to the titles in the German language.
- English needs improvement, namely ( I only present some examples...the authors should review other english mistakes):
- Line 22 and 27: “ methodologies/methods” – choose one expression, but not both
- Line 34: you need to initially write the acronym IT in full
- Keywords: You do not need to write the abbreviations MCA and LCA, because you will be repeating concepts already written in the Key words
- line 46: instead of “ its’” it should be “its”
Author Response
Dear reviewer
Please refer to the attached document in which we describe point by point how we implemented your feedback.
Thank you.

Reviewer 2 Report
Dear Authors,
I've read your manuscript on the framwork development to assess the impact of digitalisation in the energy sector with great interest. The manuscript is well structured, clearly argumented and based on robust research. The combination of three different methods (technology assessment, cost benefit analysis, multicriteria analysis, life cycle assessment and expert interviews) is suitable to develop the framework.
Author Response
Dear reviewer
Please refer to the attached document for a full overview of implemented changes.
Thank you.

This manuscript is a resubmission of an earlier submission. The following is a list of the peer review reports and author responses from that submission.
Round 1
Reviewer 1 Report
The manuscript presents mix-method to evaluate digital applications in energy sectors consisting of multi-criteria analysis, life cycle assessment and expert interviews. As presented on the manuscript, the MCA and LCA stands as separate methods with no explicit synergies. The authors have attempted to translate LCA results into MCA in 6.2.3 but this has to be further illustrated.
Figure 2: Correct typos endo - end
Line 500-501, 733 : Correct referencing errors
Author Response
Dear reviewer,
Thank you very much for your feedback.
I believe I have improved the paper regarding all the aspects you identified. In particular, the discussion on of how LCA results are translated into the MCA evaluation improves the quality of the paper. However, please note that this step cannot be completely objectified but will always remain a point where the user of the evaluation methodology has to take a subjective decision. This is now clearly identified and described as a potential weakness of the proposed combination of methodologies. Overall, the following changes based on your feedback are made:
- Revised translation of LCA into MCA results. The theory is described in Section 3 and the application in Section 4.3
- Figure 2: Typo corrected
- Line 500-501, 733 : Referencing errors corrected
Further improved points:
- Manuscript structure revised and chapter titles reworded. New structure is easier to understand and shorter titles improve readability.
- The literature review is extended, especially with a focus on the combination of evaluation methodologies. Based on this review a gap is identified and described. The properties of the proposed combination of methodologies is believed to close this gap. The required properties for evaluation methodologies for digital applications are detailed.
- Definitions moved to Appendix
- Improved separation of assumptions form discussion of results.
- Sensitivity analysis split into 2 figures, to separate the sensitivity analysis of the energy saving assumption from further influencing factors. The further influencing factors are not connected to each other in any way, thus the points in the graph are not connected with a line either.
- Added LCA result overview figure, Figure 4
- All referencing lump sums avoided
Reviewer 2 Report
This study presents a methodology for a holistic evaluation of digital applications in the energy sector to create the required transparency on benefits, obstacles and risks. The methodology is intended to create a basis for societal and political discussions about pros and cons of digital applications as well as to deliver the required information for a sustainable development and implementation of digital applications. I have the following concern.
- This paper has a serious lack of literature review with a critical analysis. The authors need to work on that part. The research gap should also clearly mentioned.
- The contribution of the paper needs to be highlighted after finding the research gap.
- The definitions of the terminologies should not be presented in the format. If you want, you can keep in the Appendix.
- Avoid using lump sum references, such as [30]–[33].
- Be concise about titles, subtitles, etc. , such as "4. Suggested evaluation methodology for digital applications in the energy sector." Please modify the entire manuscript structure for better readability.
- It is difficult to read the words in the figures.
- Appropriate references to support statements are missing throughout the manuscript.
- The writing quality of the manuscript should be improved.
Author Response
Dear reviewer,
Thank you very much for your feedback.
I believe I have improved the paper regarding all the aspects you identified. In particular, the extension of the literature review, the identification of the research gap and the description how this paper contributes to closing the gap improves the quality of the paper. Overall, the following changes based on your feedback are made:
- The literature review is extended, especially with a focus on the combination of evaluation methodologies. Based on this review a gap is identified and described. The properties of the proposed combination of methodologies is believed to close this gap. The required properties for evaluation methodologies for digital applications are detailed.
- See 1)
- Definitions moved to Appendix
- All referencing lump sums avoided
- Manuscript structure revised and chapter titles reworded. New structure is easier to understand and shorter titles improve readability.
- Improved readability of figures 1, 5 and 6 and Table 1
- Several missing references added.
- Writing and readability improved throughout the manuscript.
Further improved points:
- Improved separation of assumptions form discussion of results.
- Sensitivity analysis split into 2 figures, to separate the sensitivity analysis of the energy saving assumption from further influencing factors. The further influencing factors are not connected to each other in any way, thus the points in the graph are not connected with a line either.
- Added LCA result overview figure, Figure 4
- Revised translation of LCA into MCA results. The theory is described in Section 3 and the application in Section 4.3
Round 2
Reviewer 1 Report
Thank you for the revised version.
Author Response
Dear reviewer,
some minor changes were still made:
In lines 335-340 a new paragraph is added stating clearly, that the characteristics of the proposed method close the identified gap among evaluation methodologies.
The gap and the contribution of closing the gap is now also included in the abstract, lines 25-26 and the introduction, lines 70-79.
Thanks.
Kind regards,
Paul
Reviewer 2 Report
Please provide a clear response for each of the comment (side by side). I would suggest using a separate paper for the response letter. The contribution of the work should be written after finding the knowledge gap in the existing literature.
Author Response
Dear reviewer,
please refer to the attached document.
Thanks.

Round 3
Reviewer 2 Report
The authors have not clearly addressed the previous comments.